# Modified Target Angle as a Predictor of Success in Strabismus Management after Orbital Fracture

**DOI:** 10.3390/jcm11020287

**Published:** 2022-01-06

**Authors:** Chih-Kang Hsu, Meng-Wei Hsieh, Hsu-Chieh Chang, Yi-Hao Chen, Ke-Hung Chien

**Affiliations:** 1Department of Ophthalmology, Tri-Service General Hospital Songshan Branch, Songshan, Taipei 105, Taiwan; chikanghsu@gmail.com; 2Department of Ophthalmology, Tri-Service General Hospital, National Defense Medical Center, Taipei 114, Taiwan; doc30879@mail.ndmctsgh.edu.tw; 3Department of Ophthalmology, Taoyuan Armed Forces General Hospital, Taoyuan 325, Taiwan; nsinkoph0311@gmail.com; 4Department of Nursing, Tri-Service General Hospital, National Defense Medical Center, Taipei 114, Taiwan; n3197001@gmail.com; 5Graduate Institute of Nursing, College of Nursing, Taipei Medical University, Taipei 110, Taiwan

**Keywords:** orbital wall fracture, strabismus, diplopia, incomitant strabismus

## Abstract

Surgery for strabismus secondary to orbital fracture reconstruction surgery has had low success rates and high reoperation rates due to its incomitant nature and complex underlying mechanisms. There has been no consensus as to which of the various methods for improving the surgical results are best. We proposed a modified target angle criteria that combined the regular target angle and a favorable Hess area ratio percentage (HAR%) threshold to evaluate surgical results within the first postoperative week and conducted a retrospective chart review. According to the criteria of the modified target angle at the first postoperative week, a total of 63 patients were divided into two groups: Group 1, patients who fulfilled the criteria (49 patients); and Group 2, those who did not (14 patients). Sex, type of fracture, and the use of porous polyethylene sheets and titanium mesh during reconstruction surgery were significantly different between the groups. Group 1 showed a significantly higher percentage of patients who met the criteria of HAR% > 65% at the first week and >85% (i.e., a successful outcome) at the 6-month visit (*p* < 0.01). Additionally, Group 1 had a higher HAR% at the first postoperative week (*p* < 0.01). In conclusion, the patients meeting the criteria of the modified target angle at the first postoperative week had a favorable outcome at the 6-month visit in both ocular alignment and ocular movement.

## 1. Introduction

The primary surgical goal of orbital fracture reconstruction is to restore the orbital structure. However, there is a high incidence of surgical complication after orbital reconstruction surgeries, such as diplopia, enophthalmos lid malposition, and even optic nerve injury [1]. A total of 37–52% of patients experience diplopia after their reconstruction surgery [2,3]. Residual diplopia after orbital reconstruction surgery is reported to be the most common complication and is related to the timing of the reconstruction surgery and the extent of the fracture site [4,5,6]. Diplopia may be the result of a number of different mechanisms, but it can generally be classified as resulting from neurological paresis or a mechanical restriction [7]. Although diplopia and ocular movement disturbance can improve spontaneously over time, there is still a high incidence of residual diplopia, up to nearly 30%, requiring further management [8]. However, different from the principle of volume restoration in orbital reconstruction, treatments for consecutive strabismus-related diplopia are more challenging due to the incomitant nature and underlying mechanisms of the condition.

There is currently no consensus on the proper management of consecutive strabismus in these patients; typically, surgeons compromise on achieving a diplopia-free field in the central and down gazes [8]. Both the misalignment in the functional vision field and limitations of ocular movement that result from this procedure can be unsatisfactory to the patients. Several principles have been proposed for treating consecutive strabismus after orbital reconstruction according to its muscle pathologies and eye motility limitations [9,10,11,12,13,14,15,16]. Owing to the incomitant nature of consecutive strabismus, the Faden procedure and adjustable suture are the two most commonly applied procedures [8,17]. Recently, early postoperative deviation was found to be associated with a higher success rate and low recurrence rate in comitant strabismus [18,19]; however, few studies have compared the success rate in this condition. The binocular single vision (BSV) field is a useful parameter for measuring the functional visual area in the absence of diplopia. The Hess area ratio percentage (HAR%) compares the Hess area in the injured eye to that in the healthy eye to simulate the BSV field [20,21]. Among these patients, a HAR% > 65% is a favorable outcome indicator for the diplopic eye, and any eye movement limitations will mostly resolve thereafter [22]. However, there is no agreement on the parameters for evaluating the surgical result in the early postoperative period.

In this study, we proposed a modified target angle by combining the target angle and a HAR% threshold >65% within the first postoperative week as a parameter for evaluating early surgical results and the final success rate at the 6-month visit in patients who suffer from consecutive strabismus after orbital reconstruction surgery for orbital fractures.

## 2. Materials and Methods

This is a retrospective study on patients who underwent orbital wall fracture repair at Tri-Service General Hospital between January 2011 and December 2019 and were diagnosed with strabismus in subsequent visits. The protocol and related documents were reviewed and approved by the institutional review board of the Tri-Service General Hospital, Taipei, Taiwan. The requirement for informed consent was waived by the review board due to the retrospective basis of the study. The study was conducted under the Good Clinical Practice guidelines of Taiwan and the Declaration of Helsinki, 1964.

Patients were eligible for inclusion in the study if they had undergone orbital reconstruction surgery for orbital wall fractures and presented with persistent diplopia in subsequent visits. Patients were excluded from the study if they had other concurrent ocular diseases that might result in reduced vision prior to their orbital trauma such as amblyopia, glaucoma, cataracts, age-related macular degeneration, and diabetic retinopathy, or if they had follow-up sessions less than 6 months after their last strabismus operation.

The surgical protocol for consecutive strabismus after orbital wall fracture performed in Tri-Service General Hospital was generally followed according to the existing literature (Figure 1). Briefly, the surgery was individualized according to the patient’s examinations and intraoperative findings by combining the Faden procedure (posterior fixation procedure) and suture adjustment. Suture adjustment was performed within the first postoperative week for patients who required the procedure. If patients were diplopic in the primary position, regular recession and/or resection was added to the surgery to correct the misalignment. If diplopia was diagnosed mainly as a muscle paresis (either medial rectus or inferior rectus), a paresis-counter-paresis rule was typically applied; for example, the contralateral inferior rectus muscle was recessed in the ipsilateral inferior rectus paresis patients. If diplopia was diagnosed mainly as a muscle restriction, removal of any factor that could produce the restriction was performed first, and then regular surgery was performed; for example, in patients with inferior rectus restriction, the Faden procedure was applied for the contralateral superior rectus. In patients with mixed paresis and restriction, adhesion removal was performed first, and then the Faden procedure with or without regular recession was performed on the contralateral agonist muscle.

Recession, resection, and the Faden procedure were all performed in a forniceal approach with 6–0 Vicryl. Surgical doses were determined by the surgeon’s preference with adjustable sutures placed during the surgery. Conjunctival wound closure was performed with 8–0 Vicryl.

The following parameters were collected from the charts for the study: age at the time of operation, sex, best-corrected visual acuity, type and measurement of strabismus, surgery type for the strabismus, surgical complications, history of orbital reconstruction and strabismus operation, and alignment within the first postoperative week and at the 1-month, 3-month, and 6-month follow-up visits. Complications and reoperations, if any, were also collected during the following period.

Outcome measures were recorded from the charts as primary and secondary outcomes. (1) For the primary outcome, the angle of deviation measured within the first postoperative week was recorded as the “modified target angle” and classified as within the target range or not. The target range was referenced from Mireskandari et al. [18] for preoperative esotropia (within 8 prism diopters (PD) of esotropia) and vertical misalignment (within 4 PD of orthotropia). In the modification for assessing the treatment for strabismus from orbital fracture, we added a HAR% ≥ 65% (approximately 10″ eye movement) as a criterion for the target range. In current study, we divided patients into two subgroups according to whether they met the criteria of the modified target range. Patients who met the criteria were classified as Group 1, while who did not meet the criteria were classified as Group 2. (2) For the secondary outcome, the success of surgery was defined as an angle of deviation less than 10 PD in either the horizonal or vertical dimension in addition to at least an 85% HAR% at the 6-month follow-up visit.

The data were analyzed using SPSS software version 18.0 for Windows (SPSS Inc., Chicago, IL, USA). All the data for each group are presented as the mean (standard deviation, SD) or percentage. A value of *p* < 0.05 was considered statistically significant.

## 3. Results

Sixty subjects who suffered persistent diplopia after orbital wall fracture underwent strabismus correction in Tri-Service General Hospital between January 2011 and December 2020. Among the patients, there were 47 males (74.6%) and 16 females (25.4%), with an average age of 41.53 ± 13.62 years and a mean follow-up of 15.14 months (SD 9.17). Fifty-three patients (84.1%) suffered from an orbital fracture with an orbital floor component, 38 patients (60.3%) with a medial wall component, and 12 patients (19.0%) with an orbital rim involvement. Thirty-four patients (54.0%) had only one wall fracture (either medial or inferior wall), and 29 patients (46%) suffered from more than one wall fracture. Among the materials used for orbital reconstruction, pre-bent titanium mesh was most common and used in 41 patients (65.1%) while polypropylene sheets were used in 11 patients (17.5%), titanium mesh was used in 9 patients (14.3%), and bone grafts were used in 2 patients (3.2%) (Table 1).

To further evaluate the role of the modified target range in the first postoperative week and its impact on surgical success, we divided patients into two subgroups according to whether they met the criteria of the modified target range. As a result, 49 patients (77.8%) who met the criteria were classified as Group 1 while 14 patients (22.2%) who did not meet the criteria were classified as Group 2. Group 1 consisted of 38 male patients (77.6%) and 11 female patients (22.4%), with a mean age (SD) of 40.74 (15.23) years and a mean (SD) follow-up of 15.22 (9.17) months. In Group 2, there were 9 male patients (64.3%) and 5 female patients (35.7%) with a mean age (SD) of 42.06 (17.45) years and a mean (SD) follow-up of 14.75 (8.63) months. None of the baseline characteristics except sex were significantly different between the groups (Table 1).

Regarding the locations of the orbital fractures, 39 Group 1 patients (79.6%) had orbital floor fractures, 28 (57.1%) had medial wall fractures, and 3 (6.1%) had orbital rim fractures, while 14 patients (100%) in Group 2 had orbital floor fractures, 10 (71.4%) had medial wall fractures, and 9 (64.3%) had orbital rim fractures. All parameters associated with the location of the fractures were significantly different between groups (Table 1). Regarding the materials used for the reconstruction, in Group 1, porous polyethylene sheets (Medpor^®^, Stryker, Kalamazoo, MI, USA) were used for 10 patients (20.4%), titanium mesh was used for 5 patients (10.2%), pre-bent titanium mesh was used for 32 patients (65.3%), and autologous bone grafts were used for 2 patients (4.1%). In Group 2, one patient (7.1%) was treated with porous polyethylene sheets, 4 patients (28.6%) with titanium mesh, and 9 patients (64.3%) with pre-bent titanium mesh; no patients were treated with bone grafts in the reconstruction surgery. There was a significant difference between groups in the use of porous polyethylene sheets and titanium mesh in the reconstruction surgery. The mean time period from the orbital trauma event to the orbital reconstruction surgery was 17.36 days (SD 5.12) in Group 1 and 19.01 days (SD 6.44) in Group 2, but the difference was not significant (*p* = 1.02) (Table 1).

Regarding the information concerning strabismus after orbital fracture reconstruction, we reported patient statuses with the major type of misalignment (i.e., any component larger than 5 PD) at the primary position as most patients presented with strabismus with mixed horizontal and vertical components. As a result, there were 17 patients (34.7%) in Group 1 and 7 patients (50%) in Group 2 who presented with orthophoria in the primary position (*p* = 0.06). Among the patients with misalignment, 15 patients (30.6%) had vertical strabismus, 3 (6.1%) had horizontal strabismus, and 14 (28.6%) had mixed strabismus in Group 1, while 1 patient (7.1%) had vertical strabismus, none had horizontal strabismus, and 6 (42.9%) had mixed strabismus in Group 2. Only the number of patients with vertical misalignment was significantly different between groups (*p* = 0.03). Regarding the cause of strabismus after orbital reconstruction, 7 patients (14.3%) presented with a paralytic muscle, 12 (24.5%) with restriction and 30 (61.2%) with a mixed cause in Group 1. In Group 2, two patients experienced strabismus due to paresis, three more due to muscle restriction, and another nine due to mixed causes. The differences between groups regarding the cause of the strabismus were not significant. The mean time between orbital reconstruction surgeries and strabismus surgery interventions was 4.72 months (SD 1.54) in Group 1 and 4.63 months (SD 1.79) in Group 2, but the difference between groups was not significant (*p* = 1.41) (Table 2).

To evaluate surgical efficacy in the early postoperative period, we proposed a modified target range by combining the target angle with the HAR% during the first postoperative week. According to our study results, all patients showed greatly improved ocular motility from a mean preoperative HAR% of 47.17% (SD 27.18) to 73.03% (SD 15.59) within the first postoperative week. In the preoperative evaluation, the mean HAR% was 47.36% (SD 26.68) in Group 1 and 43.77% (SD 27.12) in Group 2, but the intergroup difference was not significant (*p* = 0.13). However, there was a significant difference in the HAR% evaluation at the first postoperative week, 74.11% (SD 14.76) in Group 1 and 65.31% (SD 17.21) in Group 2 (*p* < 0.01). Adjustable sutures were prepared for all patients in this study, and the adjustment was performed within the first postoperative week, if needed. A total of 87.8% patients (43/49) in Group 1 and 92.9% (13/14) in Group 2 required suture adjustments, but the difference was not significant (*p* = 0.11). In the evaluation of success at the 6-month postoperative visit, 38 patients (77.6%) in Group 1 and 5 patients (35.7%) in Group 2 met the criteria of success; therefore, the difference was significant (*p* < 0.01) (Table 3).

In the evaluation of surgical complications, there were no recorded complications such as slipped muscle, wound infection, granuloma, or vision loss after operations recorded in either group.

## 4. Discussion

In the management of consecutive strabismus after orbital fracture reconstruction, up to 43% of patients experienced diplopia despite strabismus correction [8]. However, there was no defined goal for assessing surgical results in the early postoperative period.

In this study, we proposed a modified target angle consisting of a previously described target angle with a HAR% range at the postoperative first week as a criterion for determining a favorable outcome in patients with secondary strabismus after orbital fracture reconstruction. From our study results, among 63 participants, 49 (77%, Group 1) met the criteria at the first postoperative week. As compared with Group 2 whose members did not meet the criteria, they had a significantly higher success rate (91.8% vs. 35.7%). We believe that the modified target angle could be applied as a clinical reference for adjustment or reoperation in these patients.

Adjustable sutures were first introduced in 1977 [23], and today are widely applied in a variety of strabismus surgeries. Zhang et al. performed a study in a large cohort of 491 adults, in which suture adjustment significantly increased the success rate to 74.8% in the adjustment group versus 61.3% in the non-adjustment group. The study showed nonsignificant differences in primary strabismus and thyroid-associated strabismus between the groups [24]. However, the efficacy of suture adjustment remains under debate; a recent study in adults with horizontal strabismus showed no significant difference in the success rate between those who received an adjustment (61.7%) and those who did not (60.3%) at early postoperative follow-ups [25]. In addition, another study in adults with non-thyroid incomitant strabismus disclosed a similar success rate between adjustable and non-adjustable suture groups (81% vs. 88%, *p* = 0.35) [26]. Another study comparing the reoperation rate for these two procedures found that adjustable sutures were associated with a significantly lower reoperation rate (5.8% in the adjustable suture group and 7.8% in the conventional group, *p* = 0.02) in horizontal strabismus but not in vertical strabismus [27]. Our study enrolled patients with consecutive strabismus after orbital reconstruction surgery. All strabismus surgeries were completed by a single surgeon (i.e., Ke-Hung Chien, M.D.), who prepared all cases with adjustable sutures due to the incomitant nature of the strabismus type in this cohort. Both groups had high rates of adjustments within the first postoperative week, but the difference was not significant (87.8% in Group 1 and 92.9% in Group 2, *p* = 0.11). Our study agreed with the prior conclusion from Zheng et al. [24], who found that adjustable sutures helped in treating incomitant strabismus and had an acceptable rate of 79.4% for the whole group. However, we did not determine the success rate for patients who did not have adjustable sutures. A future study is needed to study the application of adjustable sutures in these patients.

The definition of success in strabismus surgery varies depending on the study and the patient group. The most accepted success criterion across horizontal strabismus studies is 8–10 PD misalignment at either the 3-month, 6-month, or final follow-up [19,25,28,29,30,31,32,33]. In consecutive strabismus after orbital fracture reconstruction surgery, few studies have achieved a consensus on postoperative success criteria. Xia et al. reported their experience in this field after first setting the success criteria as less than 10 PD in horizontal deviation and less than 5 PD in vertical deviation without diplopia or head turning at the end of follow-up [14]. The authors ultimately achieved a 55.2% (i.e., 16/19) success rate by at least the 6-month follow-up [14]. Due to the concern of vision deterioration after trauma, diplopia or head tilt may not perfectly reflect misalignment after strabismus correction. A BSV parameter, such as the HAR%, would be suited to represent the status of ocular motility. Nishida et al. reported their experience in the use of the 15″ Hess screen test; patients with eye movements larger than at least 10″ (HAR% approximately 64%) were free of diplopia at the end of the follow-up period [22]. To further evaluate the surgical efficacy for the BSV field, we added a HAR% > 65% (i.e., 10″ in the Hess screen) as a criterion in the early postoperative period. In addition, Furuta et al. reported a HAR% larger than 85% as a favorable outcome in this patient group since most of them experienced no diplopia [34]. We applied this threshold at the 6-month visit as one of the criteria of success. Group 1 demonstrated a significantly higher percentage of patients who met the HAR% criterion at the first postoperative week (83.7% in Group 1 vs. 64.3% in Group 2, *p* < 0.01) and at the 6-month follow-up (79.6% in Group 1 vs. 50.0% in Group 2, *p* < 0.01). Consistent with the previous studies, the patients in our study had a final better ocular motility outcome once they achieved the early HAR% criterion.

Regarding the limitation of ocular motility secondary to orbital reconstructions, spontaneous improvement can be observed within 3–6 months [8,21]. Liu et al. reported that ocular motility significantly improved between measurements at 1 and 3 months but not between measurements at 3 and 6 months [21]. In another study by Loba et al., the authors assessed their patients after 6–12 months (mean 10.5 months) and found that 15% of patients demonstrated spontaneous resolution of their diplopia, 43.4% were not bothered by the condition, and only 28.3% needed further management [8]. Postoperative drift is another concern in strabismus surgeries. In the setting of consecutive strabismus after orbital reconstruction, its role can even be more important in the management of these patients. A study of comitant strabismus showed a positive correlation in the surgical results between the first postoperative week and 6-month follow-up visits [35]. The study results suggest the importance of achieving good ocular alignment at the postoperative first week regardless of the approach used. Mireskandari et al. proposed the idea of the target angle for evaluating the early surgical result and found that patients who met the target angle within the first postoperative week were more likely to have favorable long-term results [18,19]. In our study, we assessed the early surgical outcome by applying the target angle within the first postoperative week and the final outcome at the 6-month postoperative visit to increase the objectivity and precision of the evaluation. As a result, patients who met the modified target angle criterion (Group 1) had a significantly higher percentage of final success than those who did not (Group 2).

Management of strabismus after orbital reconstruction can be a challenging issue for strabologists not only due to its incomitant nature but also due to the complex underlying pathological mechanisms. At least six such mechanisms have been previously proposed: (1) direct damage to the extraocular muscle by a traumatic event [36], (2) muscle ischemia from the intraorbital pressure [37], (3) iatrogenic muscle damage during the reconstruction surgery [38], (4) adhesion between the muscle and nearby soft tissue or reconstructive material [11], (5) fibrosis [39], (6) entrapment by reconstructive material placed during reconstruction surgery [38], and (7) a combination of one or more of the above mechanisms [7]. Hence, there are a variety of methods to correct strabismus in these patients to alleviate diplopia in certain gazes and broaden the BSV field [8,40]. However, no consensus has been reached in the management of strabismus in this patient group; often the decision is left to the surgeon’s preferences. In this study, most surgeries were performed by a single surgeon (i.e., Ke-Hung Chien, M.D.) according to the protocol proposed in Figure 1. In some patients, additional procedures were applied during the strabismus surgeries to improve the surgical outcome; for example, we performed a simultaneous recession of the ipsilateral superior rectus muscle and inferior rectus muscle in one patient, as originally proposed by Kushner et al., and that was useful to broaden the BSV field [16]. Another helpful procedure was done in four patients with amniotic membrane application, which was well demonstrated by Strube et al. to prevent adhesion recurrence [41]. By combining the Faden procedure, adjustable sutures, and additional procedures to achieve the modified target angle in the early postoperative period, we achieved a better outcome in our study.

There are some limitations in this study. First, our study was designed to compare subjects based upon proposed criteria on a retrospective basis. Due to the application of strict inclusion criteria, only patients with a postoperative follow-up period of at least 6 months were chosen, so the study was limited by the low number of patients. Therefore, some information may have been overlooked or lost in subjects who underwent similar surgeries but did not meet the criteria. Second, the modified target angle criteria proposed in this study may not be followed by other surgeons in individual surgeries. Therefore, selection bias cannot be ignored, but a randomized controlled trial (RCT) would shed light on this issue in the future.

## 5. Conclusions

In conclusion, we proposed a modified target angle to evaluate early surgical results in patients who underwent strabismus surgery after orbital reconstruction for orbital fractures. A higher proportion of patients who met the criteria at the first postoperative week had both a wider HAR% at the first week and a higher percentage of success at the 6-month follow-up. We believe our study results could be a reference for ophthalmologists in managing patients with this disease.

## Figures and Tables

**Figure 1 jcm-11-00287-f001:**
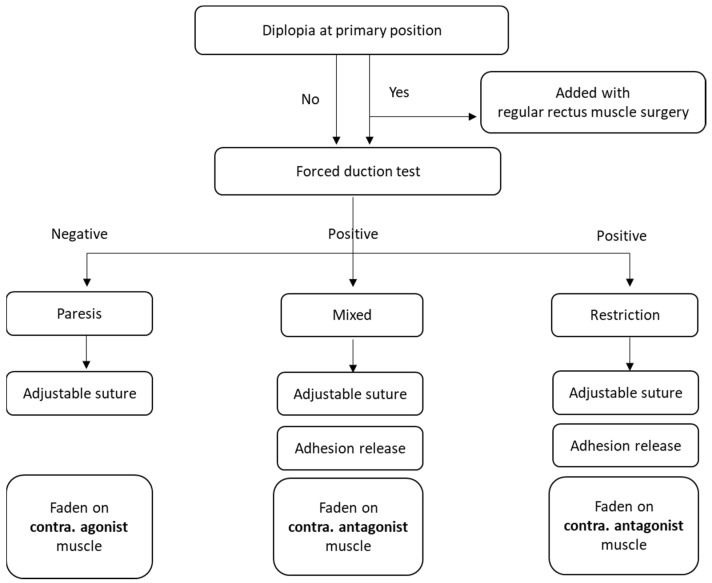
Protocol for the management of consecutive strabismus after orbital reconstruction for orbital fractures. Briefly, patients were tested to determine whether diplopia was present at the primary position, and regular rectus muscle surgery was performed to correct misalignment at the primary position if needed. A forced duction test was then performed for all patients on an outpatient basis to obtain a temporary diagnosis and in the operating room to obtain the final diagnosis. The subsequent surgical strategy was designed according to the pathological mechanism of the consecutive strabismus.

**Table 1 jcm-11-00287-t001:** Demographic characteristics of patients in the study.

	Whole Group	Group 1	Group 2	*p*
Subjects (N) (%)	63 (100%)	49 (77.8%)	14 (22.2%)	<0.01 *
Male (N) (%)	47 (74.6%)	38 (77.6%)	9 (64.3%)	<0.01 *
Female (N) (%)	16 (25.4%)	11 (22.4%)	5 (35.7%)	
Age (years) (mean) (SD)	41.53 (13.62)	40.74 (15.23)	42.06 (17.45)	1.32
Follow-up period (months) (mean) (SD)	15.14 (9.17)	15.22 (9.17)	14.75 (8.63)	1.14
Type of orbital fracture (N) (%)				
Orbital floor	53 (84.1%)	39 (79.6%)	14 (100%)	<0.01 *
Orbital medial wall	38 (60.3%)	28 (57.1%)	10 (71.4%)	<0.01 *
Orbital rim involvement	12 (19.0%)	3 (6.1%)	9 (64.3%)	<0.01 *
Materials used in orbital reconstruction (N) (%)				
Porous polyethylene sheets	11 (17.5%)	10 (20.4%)	1 (7.1%)	<0.01 *
Titanium mesh	9 (14.3%)	5 (10.2%)	4 (28.6%)	<0.01 *
pre-bent titanium mesh	41 (65.1%)	32 (65.3%)	9 (64.3%)	1.38
Bone graft	2 (3.2%)	2 (4.1%)	0 (0%)	0.06
Time from trauma to orbital reconstruction (day) (mean) (SD)	17.77 (5.45)	17.36 (5.12)	19.01 (6.44)	1.02

*p* value obtained from a comparison of group 1 and 2. SD = standard deviation, * *p* < 0.05.

**Table 2 jcm-11-00287-t002:** Detailed preoperative patient strabismus information.

	Whole Group	Group 1	Group 2	*p*
Major strabismus type (N) (%) (Component ≥ 5 PD at primary position)				
Orthophoria	24 (38.1%)	17 (34.7%)	7 (50%)	0.06
Vertical misalignment	16 (25.4%)	15 (30.6%)	1 (7.1%)	0.03 *
Horizontal misalignment	3 (4.8%)	3 (6.1%)	0 (0%)	0.24
Mixed component	20 (31.7%)	14 (28.6%)	6 (42.9%)	0.08
Strabismus cause (N) (%)				
Paresis	9 (14.3%)	7 (14.3%)	2 (14.3%)	1.65
Restriction	15 (23.8%)	12 (24.5%)	3 (21.4%)	1.32
Mixed cause	39 (61.9%)	30 (61.2%)	9 (64.3%)	1.28
Time from reconstruction to strabismus surgery (month) (mean) (SD)	4.70 (1.60)	4.72 (1.54)	4.63 (1.79)	1.41

*p* value obtained from a comparison between groups 1 and 2. SD = standard deviation. PD = prism diopter, * *p* < 0.05.

**Table 3 jcm-11-00287-t003:** Detailed patient information after strabismus surgery.

	Whole Group	Group 1	Group 2	*p*
Preoperative HAR% (mean) (SD)	47.17 (27.19)	47.36 (26.68)	43.77 (27.12)	0.13
Postoperative HAR% at first week (mean) (SD)	73.03 (15.59)	74.11 (14.76)	65.31 (17.21)	<0.01 *
Adjustment done after strabismus surgery (N) (%)	56 (88.9%)	43 (87.8%)	13 (92.9%)	0.11
HAR% > 65% at first week (N) (%)	53 (84.1%)	41 (83.7%)	9 (64.3%)	<0.01 *
HAR% > 85% at 6-month visit (N) (%)	52 (82.5%)	39 (79.6%)	7 (50%)	<0.01 *
Success (N) (%)	50 (79.4%)	38 (77.6%)	5 (35.7%)	<0.01 *

*p* value obtained from a comparison between groups 1 and 2. SD = standard deviation. HAR% = Hess area ratio, * *p* < 0.05.

## Data Availability

The datasets used and analyzed during the current study are available from the corresponding author upon reasonable request.

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
