# Peer review of "Modified Target Angle as a Predictor of Success in Strabismus Management after Orbital Fracture"

_jcm, 2022, doi:10.3390/jcm11020287_

Round 1

Reviewer 1 Report

methods and discussion should be revised since some confusion is present about "the group who fulfilled criteria" and the one that did not 

Author Response

We would like to thank the reviewer for this comment. We added the sentences below into 6th paragraph of section (line 131-134) “materials and methods” to prevent confusion” In current study, we divided patients into two subgroups according to whether they met the criteria of the modified target range. Patients who met the criteria were classified as Group 1, while who did not meet the criteria were classified as Group 2.”.

Reviewer 2 Report

Interesting proposal from the authors, whom I congratulate for their work. I also congratulate you for the critical discussion of its limitations and I encourage you to carry out prospective studies to validate your method.

I would like to propose some changes:

References should be written after the dot and not in front.

Line 82-85: “had no other comorbid ocular diseases that could result in reduced 82 vision prior to their orbital trauma, such as amblyopia, glaucoma, cataracts, age-related 83 macular degeneration, and diabetic retinopathy”. It should be noted that they are exclusion criteria and do not duplicate information.

Author Response

References should be written after the dot and not in front.

Reply:

We would like to thank the reviewer for the comments. We have revised all the reference locations in the manuscript.

Line 82-85: “had no other comorbid ocular diseases that could result in reduced vision prior to their orbital trauma, such as amblyopia, glaucoma, cataracts, age-related macular degeneration, and diabetic retinopathy”. It should be noted that they are exclusion criteria and do not duplicate information.

 Reply:

We would like to thank the reviewer for the comments. We have revised the information of exclusion accordingly in line 83-84.

Reviewer 3 Report

Reviewer Comments:

I congratulate the authors for the great work they have presented, entitled “Modified Target Angle as a Predictor of Success in Strabismus Management after Orbital Fracture”.

However, certain modifications would be necessary so as to improve the document.

  1. In the manuscript the authors present different types of orbital fractures, it would be convenient to point out if there has been any difference in the final result according to the type of fracture.
  2. The authors cite that almost all surgeries were performed by Ke-Hung Chien according to the protocol proposed in Figure 1 and that additional procedures were applied to improve the surgical outcome. For a better understanding for the reader, the number of patients in whom any of these additional procedures were performed should be indicated.

Author Response

In the manuscript the authors present different types of orbital fractures, it would be convenient to point out if there has been any difference in the final result according to the type of fracture.

Reply:

We would like to thank the reviewer for the comments. From the view of different types of fracture, there was some differences in the result of diplopia in the primary position. More complexed fractures are more prone to result in residual diplopia. Similar results were reported in our prior publication. “Anatomic Factors Predicting Postoperative Strabismus in Orbital Wall Fracture Repair. Hsu CK, Hsieh MW, Chang HC, Tai MC, Chien KH. Sci Rep. 2019 Oct 15;9(1):14785.” While in current study, we focused on surgical results related to fracture sections (ex. floor, medial wall and rim involvement) rather than certain types of fractures.  

The authors cite that almost all surgeries were performed by Ke-Hung Chien according to the protocol proposed in Figure 1 and that additional procedures were applied to improve the surgical outcome. For a better understanding for the reader, the number of patients in whom any of these additional procedures were performed should be indicated.

 Reply:

We would like to thank the reviewer for the suggestion. We added numbers of additional procedures in the paragraphs for better understanding in line 316-319.